# Transient Exposure of Endothelial Cells to Doxorubicin Leads to Long-Lasting Vascular Endothelial Growth Factor Receptor 2 Downregulation

**DOI:** 10.3390/cells11020210

**Published:** 2022-01-08

**Authors:** Silvia Graziani, Luca Scorrano, Giovanna Pontarin

**Affiliations:** 1Department of Biology, University of Padova, 35131 Padova, Italy; silvia.graziani.1@studenti.unipd.it (S.G.); luca.scorrano@unipd.it (L.S.); 2Veneto Institute of Molecular Medicine (VIMM), 35127 Padova, Italy

**Keywords:** Doxorubicin, endothelial cells, autophagy, VEGFR2, protein synthesis

## Abstract

Doxorubicin (Dox) is an effective antineoplastic drug with serious cardiotoxic side effects that persist after drug withdrawal and can lead to heart failure. Dysregulation of vascular endothelium has been linked to the development of Dox-induced cardiotoxicity, but it is unclear whether and how transient exposure to Dox leads to long-term downregulation of Endothelial Vascular Endothelial Growth Factor Receptor type2 (VEGFR2), essential for endothelial cells function. Using an in vitro model devised to study the long-lasting effects of brief endothelial cells exposure to Dox, we show that Dox leads to sustained protein synthesis inhibition and VEGFR2 downregulation. Transient Dox treatment led to the development of long-term senescence associated with a reduction in VEGFR2 levels that persisted days after drug withdrawal. By analyzing VEGFR2 turnover, we ruled out that its downregulation was depended on Dox-induced autophagy. Conversely, Dox induced p53 expression, reduced mTOR-dependent translation, and inhibited global protein synthesis. Our data contribute to a mechanistic basis to the permanent damage caused to endothelial cells by short-term Dox treatment.

## 1. Introduction

Doxorubicin (Dox) is an effective chemotherapeutic used for the treatment of numerous cancers (leukemia, breast, gastric, and ovarian cancer). Dox therapy is associated with cardiotoxic effect that can manifest acutely as well as years after treatment has been discontinued [1]. Current strategies to prevent Dox-induced cardiotoxicity are scarce and ineffective. In former cancer patients, Dox cardiomyopathy is particularly difficult to treat and can progress to fatal heart failure. Cardiomyocytes have been long considered the main target in the progressive alterations of the cardiac muscle caused by Dox. In this classic model of Dox cardiotoxicity, cardiomyocyte mitochondrial damage plays a key role [2]. Dox accumulates in mitochondria by specifically binding to cardiolipin in the inner mitochondrial membrane and participates to several processes leading to enhanced ROS production, changes in iron metabolism, impaired mitochondrial Ca^+^^+^ homeostasis, inhibition of mitochondrial topoisomerase Iiβ, and dysregulation of autophagy. All these proposed mechanisms ultimately lead to cardiomyocyte dysfunction and death.

Other cell types have also been identified as important in the development of Dox-cardiomyopathy [3]. In particular, endothelial cells (ECs) are severely affected by Dox treatment. Dox decreases capillary density in treated mouse hearts, increases cardiac microvasculature permeability, and suppresses formation of vascular networks in human cardiac endothelial cells in vitro [4]. Apoptosis is not considered the primary mechanism of endothelial injury by Dox, but other mechanisms have been proposed, including reduced regenerative ability of EC progenitors [5], aberrant activation of TGF-β pathway [6], and impaired Atg7-dependent autophagy [7].

VEGF receptor type2 (VEGFR2) is a tyrosine kinase receptor highly expressed in ECs. Upon the binding to its VEGF-A ligand, VEGFR2 mediates signaling cascades to induce angiogenic response [8]. These signaling pathways include the activation of phospholipase C-γ-ERK1/2 pathway, PI3K-AKT-mTOR pathway, SCR kinases, and small GTPases pathways that are involved in cell survival, migration, and polarization, as well as regulation of endothelial junctions, vascular barrier, and vasomotion during vascular development. In cardiac ECs, VEGFR2 signaling can determine whether cardiac growth is physiological or pathological [9], underlining the importance of paracrine endothelium to cardiomyocytes signaling. Given the critical role of VEGFR2 signaling, VEGFR2 levels are tightly regulated at multiple levels. One important mechanism involves endocytosis and trafficking [10]. VEGFR2 activated by the binding to VEGF-A at the cell surface is rapidly internalized into early endosomes that serve as the platform for VEGFR2 signaling. A fraction of VEGFR2 in early endosomes is ubiquitinated and sorted to lysosomes for degradation, while the remaining is recycled back to the plasma membrane.

An open question is how Dox treatment can lead to deleterious heart problems years after it has been discontinued. Mechanistic in vitro studies are limited by the nature of the model systems employed, where Dox effects are mostly investigated during continuous exposure to the drug. This, however, does not adequately reproduce the long-lasting modifications that can be caused by relatively brief exposures of cells to Dox and can have their clinical counterpart in the delayed onset of cardiomyopathy, calling for the establishment of cellular models where long-term effects of Dox can be studied after drug withdrawal.

To overcome the lack of an appropriate model to study the long-lasting effects of Dox treatment on endothelial cells, we devised an in vitro system where HUVECs were pulsed with Dox and then analyzed days after drug removal. We employ this system to investigate whether a brief treatment with Dox permanently affects VEGFR2 levels. Our data indicate that transient Dox exposure leads to an enduring reduction in VEGFR2 protein levels by a mechanism involving p53/mTOR-mediated repression of global protein synthesis.

## 2. Materials and Methods

### 2.1. Cell Culture and Chemicals

Human umbilical vein endothelial cells (HUVECs) were purchased from Lonza (Cat. C2519A). Cells below passage 8 were used for experimental manipulations and grown in Endothelial Cell Growth Medium 2 (EGM-2, Cat. CC-22211, PromoCell, Heidelberg, Germany) supplemented with 2% fetal calf serum, growth factors, such as Epidermal Growth Factor (5 ng/mL), Fibroblast Growth Factor (10 ng/mL), Insulin-like Growth Factor (20 ng/mL), Vascular Endothelial Growth Factor 165 (0.5 ng/mL), ascorbic acid (1 µg/mL), and hydrocortisone (0.2 µg/mL) (Cat. C-39211, PromoCell, Heidelberg, Germany). HUVEC were cultured on 0.2% pre-coated gelatin plates in a 37 °C incubator with humidified atmosphere of 5% CO_2_. The chemicals Doxorubicin (Cat. D1515), Chloroquine (Cat. C6628), and Lactacystin (Cat. L6785) were purchased from Sigma-Aldrich (St. Louis, MO, USA). Doxorubicin and Lactacystin were dissolved in DMSO. Puromycin (Cat. A1113802) was purchased from Gibco™ (Shanghai, China).

### 2.2. Cell Death Assay

Cell death was detected by flow cytometry analysis by Annexin V-FITC/DRAQ7 staining. Cells were harvested by trypsinization and centrifuged with culture medium at 700 rcf for 5 min. Each sample was resuspended in 200 µL Binding buffer containing 2 µL Annexin V-FITC supplied by the FITC AnnexinV/Dead Cell Apoptosis Kit (Cat. V13242, Invitrogen, Waltham, MA, USA). After incubation for 15 min, DRAQ7 (Cat. D15106, Invitrogen, Eugene, OR, USA) was added to each tube (1:200 dilution) and samples were analyzed with FACSCanto II (BD Biosciences, Franklin Lakes, NJ, USA) using BD FACSDiva™ Software (BD Biosciences, Franklin Lakes, NJ, USA). For each sample were acquired at least 10,000 events.

### 2.3. β-Galactosidase Assay

The assay of SA-β-galactosidase (SA-β-gal) was performed according to the protocol of Senescent Detection Kit (Cat. QIA117, Millipore, Darmstadt, Germany). Cells were fixed in fixing solution (20% formaldehyde, 2% glutaraldehyde, 70 mM Na_2_HPO_4_, 14.7 mM KH_2_PO_4_, 1.37 mM NaCl, 26.8 mM KCl) for 10 min and incubated overnight at 37 °C with the staining mixture containing the X-gal (5-Bromo-4-chloro-3-indolyl-β-d-galactopyranoside). Blue SA-β-gal positive cells were visualized in bright field microscopy with Leica DMI4000B (Leica, Wetzlar, Germany). Counting was performed by an investigator blinded to the identity of the sample and data are expressed as the percentage of positive cells over cells counted (>100 cells/sample).

### 2.4. Real-Time Quantitative PCR

Total RNA was extracted from cells using TRIzol^®^ Reagent (Cat. 15596018, Thermo Fisher Scientific, Carlsbad, CA, USA). Isopropanol was used for RNA precipitation, after phenol/chloroform separation. Samples were dried at room temperature and dissolved in UltraPure DNAse/RNAse-Free Distilled Water (Cat. 10977-035, Invitrogen, Paisley, UK) for 10 min at 55 °C. The concentration of isolated mRNA was determined using NanoDrop ND-1000 Spectrophotometer (Thermo Fisher Scientific, Waltham, MA, USA). According to the manufacturer’s protocol, High-Capacity cDNA Reverse Transcription Kit (Cat. 4374966, Applied Biosystem, Vilnius, Lithuania) was used for cDNA synthesis starting from 1 µg RNA template. The qPCR was performed with the use of HOT FirePol EvaGreen qPCR Supermix 5× (Cat. 08-36-00001, Solis BioDyne, Tartu, Estonia) and QuantStudio 5 Real-Time PCR system (Thermo Fisher Scientific) in sequence: an initial step at 95 °C for 15 min to activate polymerase, followed by 39 cycles at 95 °C for 15 s (denaturation), 60 °C for 45 s (annealing), and 72 °C for 30 s (elongation). The primers used were specific for human: VEGFR2 (F: 5′-CCCAGATGCCGTGCATGAG-3′ and R: 5′-ATGACATTTTGATCATGGAGC-3′), VEGF-A (F:5′-TACCTCCACCATGCCAAGTG-3′ and R:5′-TCTTCCTCCTCCCGTCTTAGTA-3′), VEGFR1 (F: 5′-CCGGACTGTGGCTGTGAAAA-3′ and R: 5′-TGGCCAATGTGGGTCAAGAT-3′), FGFR1 (F: 5′-ATCACGGCTCTCCTCCAGTG and R: 5′-AAGACCAGTCTGTCCCGAGGC-3′). Tubulin (F: 5′-CTTCGTCTCCGCCATCAG-3′ and R: 5′-CGTGTTCCAGGCAGTAGAGC-3′) was used as housekeeping for normalization of examined gene expression. Each sample was analyzed in triplicate and relative mRNA expression was calculated as 2^−ΔΔCt^ (ΔCt= Ct_Gene of interest_ − Ct_Tubulin gene_). Data are normalized to not-treated cells.

### 2.5. Immunoblotting

Cells were lysed in RIPA buffer (Cat. R0278, Sigma Aldrich, St. Louis, MO, USA) added with phosphatase inhibitor (Cat. 04 906 845 001, Roche, Mannheim, Germany) and protease inhibitor cocktail (Cat. 05 892 970 001, Roche, Mannheim, Germany). After 30 min incubation on ice, the lysate was centrifuged at 19,000 rcf for 15 min at 4 °C and the supernatant was collected. Protein concentration was determined by Bradford (BIO-RAD Protein assay, Cat. 500-006, BIO-RAD, Munich, Germany). Equal amounts of proteins (15–20 μg) were resolved using a 4–12% SDS-PAGE Gel (Cat. NP0321, Invitrogen, Carlsbad, CA, USA), or a homemade 15% SDS-PAGE gel, and transferred to nitrocellulose membranes (Cat. 10600001, Amersham™, Schnelldorf Germany). Membranes were blocked with 4% BSA in 0.1% Tween 20-TBS buffer (T-TBS) for 1 h and incubated overnight at 4 °C with diluted primary antibody in 2% BSA in T-TBS. Primary antibodies used were: VEGFR2 1:3000 (Cat. 9698, Cell Signaling Technology, Danvers, MA, USA), FGFR1 1:2000 (Cat. 9740, Cell Signaling Technology, Danvers, MA, USA), 4E-BP1 1:2000 (Cat. 9452, Cell Signaling Technology, Danvers, MA, USA), p70 1:2000 (Cat. 9202, Cell Signaling Technology, Danvers, MA, USA), p-4E-BP1 (Ser^65^) 1:4000 (Cat. 9451, Cell Signaling Technology, Danvers, MA, USA), p-p70 S6K (Thr^389^) 1:1000 (Cat. 9205, Cell Signaling Technology, Danvers, MA, USA), LC3 (Cat NB100-2220, Novus Biologicals 1:1000, Centennial, CO, USA), Lamin B1 1:1,000 (Cat. Sc-6216, Santa Cruz Biotechnology, Dallas, TX, USA), p21 1:500 (Cat. Sc-817, Santa Cruz Biotechnology, Dallas, TX, USA), p53 1:2000 (Cat. Sc-263, Santa Cruz Biotechnology, 1: Dallas, TX, USA), Actin 1:4000 (Cat. A5441, Sigma Aldrich St. Louis, MO, USA), Vinculin 1:4000 (Cat. V9264, Sigma Aldrich, St. Louis, MO, USA), GAPDH 1:4000 (Cat. CB1001, Millipore, Burlington, MA, USA), Puromycin 1:10,000 (Cat. MAEBE343, Millipore, Burlington, MA, USA).

For chemiluminescence detection (ECL), membranes were incubated for 1 h with the appropriate secondary antibodies conjugated with horse radish peroxidase (HRP). Signal was developed using Immobilon^®^ Forte Western HRP Substrate (Cat. WBLUF500, Millipore, Burlington, MA, USA) and acquired using ImageQuant™ LAS 4000 mini (GE Healthcare, Chicago, IL, USA). Quantification of the signal was performed using ImageJ software (National Institutes of Health, Bethesda, MD, USA).

### 2.6. Statistical Analysis

Number of replicates are indicated in the figure legends. All graphs report mean ± SEM values of replicates. Comparison between two groups were performed by unpaired two-tailed Student’s *t* test using GraphPad Prism 8.0.2 (GraphPad Software, San Diego, CA, USA). *p* values were two-tailed and values <0.05 were considered to indicate statistical significance. *p* < 0.05, *p* < 0.01, *p* < 0.001, *p* < 0.0001 are designed in all figures with *, **, ***, ****, respectively.

## 3. Results

### 3.1. Transient Exposure of ECs to Doxorubicin Triggers Cellular Senescence That Is Sustained after Dox Removal

Dox toxicity in the clinical setting can develop even years after patients discontinued the drug. This remarkable feature complicates mechanistic studies in the lab that normally rely on acute, continuous exposure of target cells to the drug. In order to overcome this limitation and to evaluate short as well as long term effects of endothelial cells exposure to Dox, we setup an in vitro system where HUVECs were treated with 250 nM Dox, a concentration commonly obtained in clinical use for 24 h, followed by extensive washing and culturing in drug-free complete medium for several additional days. Twenty-four hours after exposure to Dox, we recorded an increase in the amount of HUVECs apoptosis. This same level of cell death was measured eight days after Dox treatment, indicating that apoptosis is an acute effect of Dox treatment (Figure 1A). Conversely, senescence, measured by the fraction of SA-β-gal positive cells, increased over time (Figure 1B). The development of a time-dependent senescent phenotype was confirmed by the quasi-complete disappearance of Cyclin A and Lamin B1, two markers of cell proliferation (Figure 1C). The cell cycle arrest was assessed by the appearance of the CDK inhibitor p21^Waf/Cip1^.

Cellular senescence is characterized by an irreversible cell cycle blockage independently of the presence of growth factors. In ECs, the proliferation is mediated by the Vascular Endothelial Growth Factor A (VEGF-A), whose expression is significantly increased in the presence of the drug, while during senescence was similar to the not-treated cells (Figure 1E). Considering these data and the constant presence of VEGF-A in the culture medium, we concluded that in our endothelial cellular model, VEGF-A did not avoid the block of cell cycle and the induction of senescence.

In sum, Dox-treated EC undergo senescence in a time-dependent manner that progresses after removal of the drug.

### 3.2. Transient Exposure of ECs to Doxorubicin Leads to Reduced VEGFR2 Level

VEGFR2 is a key mediator of EC survival, activation, and proliferation and the inhibition of VEGFR2 signaling leads to senescence [11]. We therefore verified if the observed Dox-induced senescence was associated with lower VEGFR2 levels. First, we noticed that 24 h after Dox treatment, total VEGFR2 levels were almost nil and that even eight days after Dox removal they were reduced by ~75% (Figure 1C,D). These results indicate that VEGFR2 levels are extremely sensitive to Dox and call for a more detailed time course analysis of the impact of Dox on VEGFR2 expression. We therefore measured VEGFR2 protein levels early after Dox treatment. The receptor levels halved within 6 h and then decreased at a much slower pace (Figure 2A,B). Similarly, VEGFR2 mRNA levels significantly dropped at 6 h, then reverting to the values retrieved in untreated cells (Figure 2C). VEGFR2 mRNA levels were similarly unaffected eight days after Dox removal (Appendix A), when VEGFR2 protein levels were, however, decreased, suggesting the existence of post transcriptional mechanism that contribute to downregulate VEGFR2 levels in the late stages following Dox removal and calling for a deeper investigation of the link between Dox exposure and delayed VEGFR2 downregulation.

In addition, we considered FGFR1 and VEGFR1 as additional growth factor receptors, whose expression could be impaired by Dox treatment (Appendix A). The mRNA levels were not affected by the presence of the drug and during Dox-induced senescence. FGFR1 protein level decreased during Dox treatment similarly to VEGFR2, while it turned back to the initial level after the removal of the drug during senescence. These data underlined that only VEGFR2 was permanently affected by Dox.

### 3.3. Doxorubicin Enhances Autophagy in ECs

We reasoned that Dox could increase VEGFR2 degradation or reduce its translation. Cycloheximide treatment revealed that in unperturbed HUVECs, VEGFR2 underwent a rapid turnover, with a similar rate of proteolysis and resynthesis (Appendix A). We therefore decided to analyze both sides of the synthesis/degradation equilibrium of VEGFR2. Because Dox affects autophagy in normal and transformed cells, we measured whether degradation of VEGFR2 depended on Dox stimulated autophagy. Notably, immunoblots for the classic autophagy marker LC3 indicated that in ECs treated with Dox, the LC3-I to LC3-II conversion was increased. Accumulation of LC3-II could result from a stimulation of basal autophagy or from decreased autophagic flux. We therefore used Chloroquine (CQ), which inhibits lysosomal acidification and hence autophagosome-lysosome fusion [12] to evaluate autophagic flux in unperturbed and Dox-treated HUVECs. These measurements indicated that autophagic flux was conversely stimulated by Dox treatment (Figure 3A,B), suggesting that Dox might reduce VEGFR2 levels by enhancing its trafficking and autophagic degradation. We therefore analyzed whether autophagy inhibition restored VEGFR2 level in Dox-treated HUVECs. However, CQ treatment did not alter the amount of the 230–200 kDa glycosylated receptor (Figure 3C,D), but led to a significant accumulation of the 160 kDa truncated form of VEGFR2 (Figure 3C,E). This fragment is generated by proteasome-mediated degradation from the full-length receptor prior to its lysosomal degradation [13] and its accumulation indicated that CQ efficiently blocked VEGFR2 lysosomal degradation. However, because CQ did not restore the amount of VEGFR2 in Dox-treated cells, we surmised that Dox-mediated reduction in VEGFR2 levels are autophagy independent. We therefore investigated if Dox decreased total VEGFR2 levels by stimulating degradation of the 160 kDa VEGFR2 fragment. The specific 26S proteasome inhibitor Lactacystin (LC) efficaciously reduced levels of the 160 kDa VEGFR2 form in both untreated and Dox treated cells. However, LC was unable to restore levels of full length VEGFR2 in Dox treated HUVECs, ruling out a role for proteasome-dependent VEGFR2 degradation in the observed Dox-induced VEGFR2 reduction.

### 3.4. Doxorubicin Inhibits Global Protein Synthesis through the mTOR Axis

Because we could not find evidence that Dox decreased VEGFR2 levels by stimulating degradation of this receptor, we turned our attention to the possibility that Dox impaired VEGFR2 protein synthesis. We pulsed HUVECs for 10 min with Puromycin and we determined the incorporation of the drug in newly synthesized proteins by immunoblotting [14]. Dox inhibited protein synthesis by 50% in HUVECs (Figure 3A,B). Dox can affect protein synthesis by impinging on the p53-mTOR axis [15]. We therefore evaluated first whether Dox treatment led to the accumulation of the tumor suppressor p53 and its target gene p21^Cip1^. Immunoblots for p53 and p21^Cip^ confirmed that they rapidly accumulated, peaking 6 h after Dox treatment (Figure 4C–E). Interestingly, p21 is a CDK inhibitor required for p53-induced cell cycle arrest and an inducer of senescence [16], lending further support to our earlier observation of Dox-induced senescence. Comforted by the finding that Dox induces p53 accumulation, we next evaluated whether Dox inhibited mTOR activation. We therefore monitored the phosphorylation status of the two mTOR downstream effectors, p70 S6K and 4EBP1, and noticed that phosphorylation of both targets was strongly reduced (Figure 4F–H). We observed a sustained inhibition of p70 S6K and 4EBP1 phosphorylation that lasted eight days following Dox pulse and withdrawal (Appendix A), suggesting that mTOR inhibition remarkably persists long after Dox treatment has been stopped. Collectively, these results suggest that in ECs, Dox represses VEGFR2 translation via inhibition of the mTORC1 pathway.

## 4. Discussion

Preservation of heart function during Dox treatment is a current unmet clinical need. While a large body of evidence points to a role for cardiomyocyte mitochondria as mediators of Dox toxicity, Dox can also directly damage heart endothelial cells that crucially control heart size and function. Most of these studies rely on analyses of Dox toxicity in models where large, non-clinical Dox concentrations are administered continuously. These models do not faithfully reproduce the clinical reality, where anthracycline cardiomyopathy develops years after drug withdrawal, calling for an in vitro system that can recapitulate the long-term effects of a brief exposure to a clinically relevant Dox concentration. With this question in mind, we devised an in vitro system to study prolonged effects of short exposure of endothelial cells to low Dox concentrations similar to those achieved in the plasma of treated patients [17]. We show that Dox has long-lasting effects on endothelial cell senescence, protein synthesis, ultimately impacting on levels of VEGFR2, the crucial receptor involved in EC activation and proliferation.

In isolated rat cardiac microvascular endothelial cells, anthracycline therapy reduces cell viability, VEGF-A release, and VEGFR2 levels [18]. However, these effects were observed at Dox concentrations >1µM, far above the Dox plasma levels measured in patients, and during acute treatments that do not mirror the clinical conditions of delayed Dox-induced heart failure. By using our carefully devised in vitro model, we found that clinically relevant Dox concentrations only slightly affected cell viability and that on the other hand they caused EC senescence days after Dox withdrawal.

Dox is known to induce EC senescence, albeit not associated to a “classic” senescence associated secretory pattern (SASP), the set of cytokines secreted by senescent cells that contribute to remodel the tissue niche and to trigger inflammation. The lack of a typical senescence secretory response in Dox-treated ECs is due to the ability of Dox to activate the mTOR pathway [19]. We similarly observed mTOR activation as a long-lasting effect of Dox treatment. We correlate mTOR activation to the ability of Dox to induce p53, a key mediator of DNA damage response that negatively regulates mTOR signaling [20]. Inhibition of mTOR curtails protein synthesis, potentially explaining how Dox treatment reduces VEGFR2 levels. Indeed, the mTOR inhibitor rapamycin suppresses tumor angiogenesis by downregulating VEGF/VEGFR2 levels through the inhibition on translation initiation [21].

In principle, VEGFR2 downregulation could have resulted from enhanced degradation or from transcriptional repression. However, our data ruled out defects in VEGFR2 transcription. As for VEGFR2 degradation, this process depends on autophagy, the inhibition of which can increase VEGFR2 levels and angiogenesis [22]. While we confirmed that Dox increased autophagic flux [23], inhibition of autophagy or of the proteasome did not restore VEGFR2 levels in Dox-treated HUVECs. These data support that lower levels of VEGFR2 in our model of delayed Dox toxicity are a consequence of the suppression of mTOR signaling and of protein synthesis. A very similar readout was recently obtained in endothelial progenitor cells treated with low doses of Dox [24].

VEGFR2 signaling is crucial for ECs’ activation and hence for vascular development and maintenance [8]. Activated cardiac ECs release paracrine signals that can modulate the growth of cardiomyocytes [9]. Not surprisingly, perturbed crosstalk between ECs and cardiomyocytes is involved in the pathogenesis of heart diseases [25]. Thus, downregulation of this receptor in heart ECs exposed to Dox concentrations might play a key role in the development of anthracycline-induced cardiomyopathy [18].

## Figures and Tables

**Figure 1 cells-11-00210-f001:**
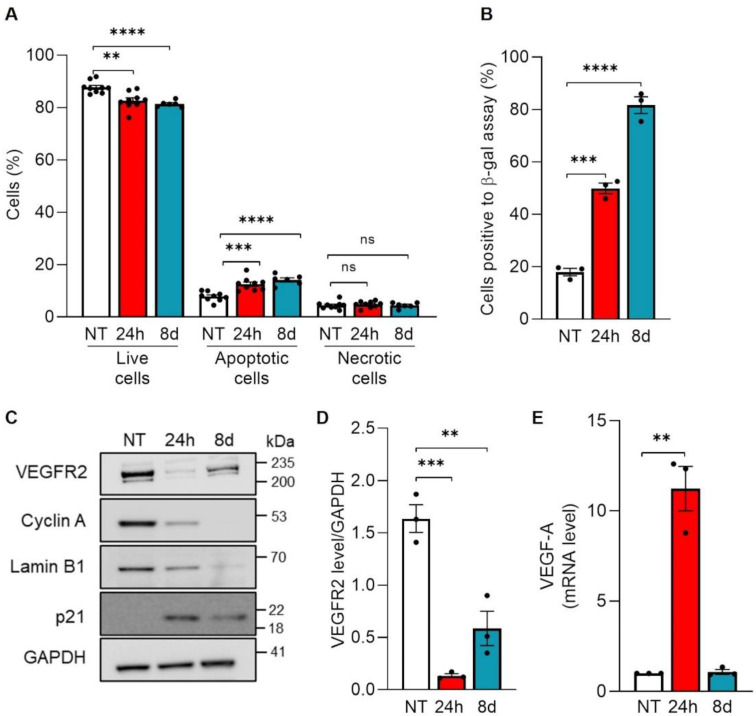
Effects of a transient exposure of endothelial cells to Doxorubicin. HUVECs were incubated with 250 nM Doxorubicin for 24 h. After extensive washing, the medium was replaced with free-drug complete medium and cells were maintained in culture for additional days. Samples were collected and analyzed at 24 h of treatment (24 h), red histogram, and after 8 days (8 d), blue histogram. NT = not treated cells, white histogram. (**A**) Percentage of live, apoptotic, and necrotic cells determined by cytofluorimetry. Data are shown as mean ± SEM of *n* = 6–9 independent experiments per condition. ns *p* >0.05, ** *p* < 0.01 *** *p* < 0.001, **** *p* < 0.0001 by unpaired Student’s *t* test. (**B**) Percentage of positive cells for Senescence-Associated β-galactosidase activity. Data are shown as mean ± SEM of *n* = 3 experiments per condition; *** *p* < 0.001, **** *p* < 0.0001 by unpaired Student’s *t* test. (**C**) Cell lysates analyzed by immunoblotting with the indicated antibodies. (**D**) Quantification of VEGFR2 protein level by densitometric analysis. Data are shown as mean ± SEM of protein level normalized to GAPDH. *n* = 3 independent experiments; ** *p* < 0.01, *** *p* < 0.001 by unpaired Student’s *t* test. (**E**) mRNA level of VEGF-A determined by qRT-PCR. Each sample was normalized to β-tubulin and data are expressed as fold increase compared to the untreated sample. Data are shown as mean ± SEM of *n* = 3 independent experiments per condition. ** *p* < 0.01.

**Figure 2 cells-11-00210-f002:**
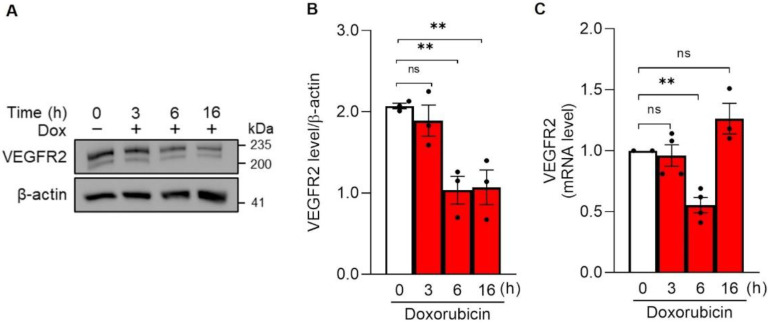
VEGFR2 protein levels decrease early following Doxorubicin treatment. HUVECs were incubated with 250 nM Doxorubicin and samples were collected at the indicated time during the treatment, red histogram. White histogram represents the not treated cells. (**A**) Representative immunoblot of HUVEC lysates analyzed using an antibody against VEGFR2. (**B**) Quantification of VEGFR2 protein level by densitometric analysis. Data are shown as mean ± SEM of protein level normalized to β-actin. *n* = 3 independent experiments per condition. ns *p* > 0.05, ** *p* < 0.01, by unpaired Student’s *t* test. (**C**) mRNA level of VEGFR2 determined by qRT-PCR. Each sample was normalized to β-tubulin and data are expressed as fold increase compared to the untreated sample. Data are shown as mean ± SEM of *n* = 3–4 independent experiments per condition. ns *p* > 0.05, ** *p* < 0.01 by unpaired Student’s *t* test.

**Figure 3 cells-11-00210-f003:**
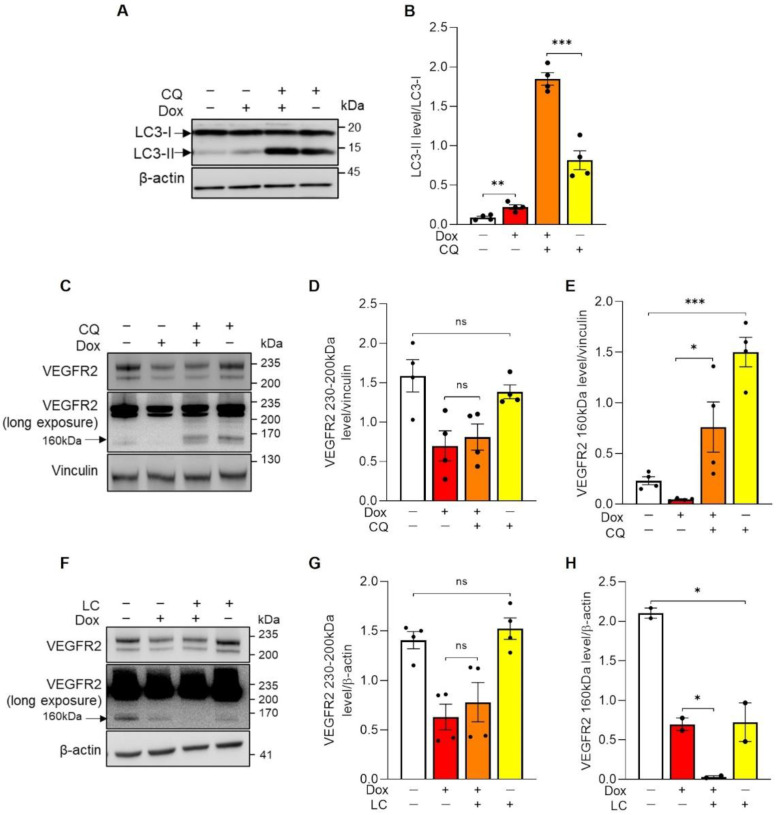
Doxorubicin-induced VEGFR2 reduction is not due to an increased degradation. (**A**–**E**) HUVECs were incubated with 250 nM Doxorubicin in absence or presence of 50 µM Chloroquine (CQ). After 16 h, cells were lysed and analyzed by immunoblotting with indicated antibodies. (**A**) Representative immunoblot. Arrows indicate the LC3-II and LC3-I forms. (**B**) Quantification of LC3 protein level by densitometric analysis. Data are shown as mean ± SEM of LC3-II level normalized to LC3-I. *n* = 4 independent experiments per condition. ** *p* < 0.01, *** *p* < 0.001 by unpaired Student’s *t* test. (**C**) Representative immunoblot. Arrow indicates the cleaved form of VEGFR2 in the same long-exposed blot. (**D**,**E**) Quantification of the mature (VEGFR2 230–200 kDa) and cleaved VEGFR2 (VEGFR2 160 kDa) protein levels by densitometric analysis. Data are shown as mean ± SEM of protein level normalized to vinculin. *n* = 4 independent experiments per condition. ns *p* > 0,05, * *p* < 0.05, *** *p* < 0.001 by unpaired Student’s *t* test. (**F**–**H**) HUVECs were incubated with 250 nM Doxorubicin in absence or presence of 5 µM Lactacystin (LC). After 16 h, cells were lysed and analyzed by immunoblotting with indicated antibodies. (**F**) Representative immunoblot. Arrow indicates the cleaved form of VEGFR2 in the same long-exposed blot. (**G**,**H**) Quantification of the mature (VEGFR2 230–200 kDa) and cleaved VEGFR2 (VEGFR2 160 kDa) forms by densitometric analysis. Data are shown as mean ± SEM of protein level normalized to β-actin. *n* = 2–4 independent experiments per condition. ns *p* > 0.05, * *p* < 0.05 by unpaired Student’s *t* test. Not treated cells = white histogram, Dox = red histogram, Dox + CQ/LC = orange histogram and CQ/LC = yellow histogram.

**Figure 4 cells-11-00210-f004:**
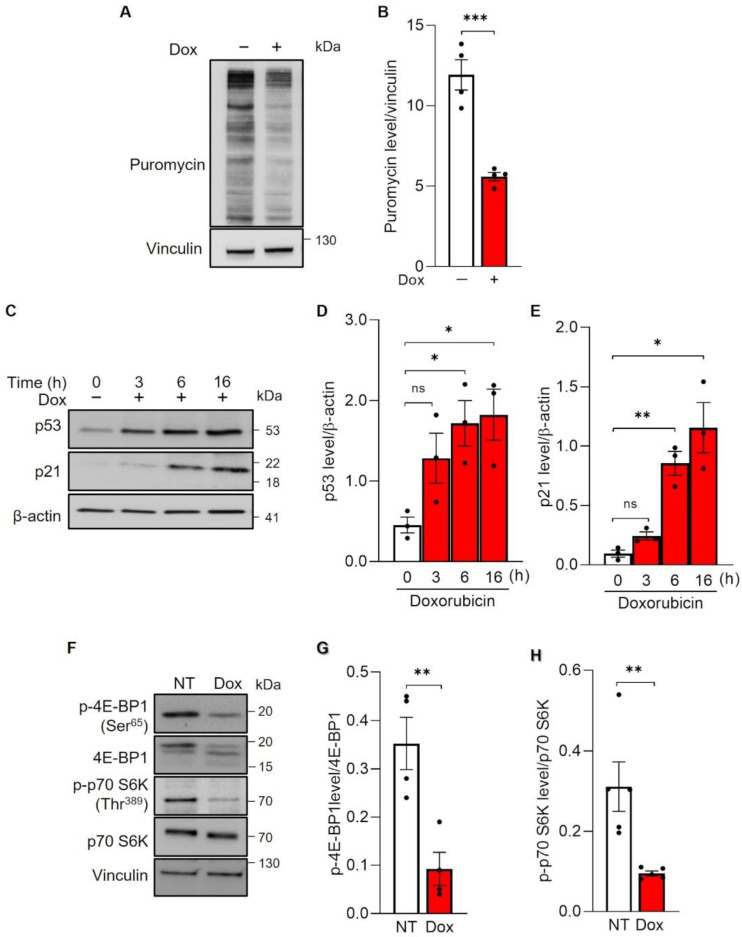
Global protein synthesis and mTOR activity during Doxorubicin treatment. (**A**,**B**) HUVECs were incubated (+), red histogram, or not (−), white histogram, with 250 nM Doxorubicin (Dox) for 16 h and then labelled for 10 min with of 90 µM Puromycin. Cells were lysed and analyzed by immunoblotting using an antibody against Puromycin. (**A**) Representatives immunoblot. (**B**) Quantification of puromycin levels by densitometric analysis. Data are shown as mean ± SEM of puromycin levels normalized to vinculin. *n* = 4 independent experiments per condition. *** *p* < 0.001 by unpaired Student’s *t* test. (**C**–**E**) HUVECs were incubated with 250 nM Doxorubicin and samples were collected at the indicated time during the treatment, red histogram. White histogram represents not-treated cells. (**C**) Representative immunoblot of HUVEC lysates analyzed using indicates antibodies. (**D**,**E**) Quantification of p53 and p21 protein levels, respectively, by densitometric analysis. Data are shown as mean ± SEM of protein levels normalized to β-actin, *n* = 3 independent experiments per condition. ns *p* > 0.05, * *p* < 0.05, ** *p* < 0.01 by unpaired Student’s *t* test. (**F**–**H**) HUVECs were treated (Dox, red histogram) or not (NT, white histogram) with 250 nM Doxorubicin for 16 h and analyzed by immunoblotting with the indicated antibodies for mTORC1 signaling. (**F**) Representative immunoblot. (**G**,**H**) Quantification of p-4E-BP1 (Ser^65^ and total) and p-p70 S6K (Thr^389^ and total), respectively, by densitometric analysis. Data are shown as mean ± SEM of phosphorylated protein levels normalized to total protein level. *n* = 4 independent experiments per condition. ** *p* < 0.01 by unpaired Student’s *t* test.

## Data Availability

The data presented in the current study are available upon request to the corresponding authors.

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
