# Peer review of "Transient Exposure of Endothelial Cells to Doxorubicin Leads to Long-Lasting Vascular Endothelial Growth Factor Receptor 2 Downregulation"

_cells, 2022, doi:10.3390/cells11020210_

Round 1

Reviewer 1 Report

This is an important and well executed study that provides detailed mechanistic understanding of Doxorubicin mediated long-lasting down-regulation of VEGFR2, (expressed mostly in vascular endothelial cells, and crucial for endothelial cells function) providing the basis to the late doxorubicin-induced cardiomyopathy. I have a minor suggestion on the use of colored bars in the figures. The authors can either specify in the legend what the colors mean or probably keep them block n white and just label appropriately.

Author Response

In the new version of the manuscript, we have specified in the figure legend the meaning of each coloured histogram.

Reviewer 2 Report

The manuscript from Graziani et al presents a study looking at the detrimental effect of doxorubicin on primary human endothelial cells focusing on the doxorubicin-mediated induction of senescence and down-regulation of VEGFR-2 levels.

The experimental data is presented well and is technically sound.  However, the main problem is that the manuscript reports on a series of separate findings such as senescence, reduced protein synthesis, reduced translation of VEGFR-2.  There needs to be more of an attempt to package this up into a more focussed study with an eye on how the findings could be used to try and overcome the doxorubicin-mediated induction of senescence. One would assume that in addition to VEGFR-2, numerous other growth factor receptors, such as FGFR-1 and HGFR, are also reduced following doxorubicin treatment? What is the role of senescence in the pathophysiological response to doxorubicin, does VEGF-A protect against the senescence with doxorubicin?

  1. Senescence is measured by increased b-galactosidase activity. Is there a concomitant induction in p16INK4A levels?
  2. Does the reduced VEGFR-2 level also lead to a concomitant reduction in VEGF-A mediated signalling such as reduced activation of Erk1/2 and reduced phosphorylation of AKT?
  3. Can incubation with VEGF-A (added prior to doxorubicin and at a concentration that robustly activates VEGFR-2 phosphorylation) protect against the doxorubicin-mediated reduction of VEGFR-2 levels and induction of senescence?
  4. The authors show that doxorubicin reduces S6K phosphorylation and 4EBP1 phosphorylation over 8 days. Can they be sure that this is not a feature of reduced protein levels as doxorubicin appears to reduce the level of 4E-BP1 and also S6K (Fig S3)?

Round 2

Reviewer 2 Report

his revised manuscript has now been improved and answers the concerns/issues with the original manuscript.